# Development of a Health Research Portfolio Based on Priority Topics for Peruvian Social Health Insurance (ESSALUD) in 2023–2025: A Collaborative Approach to Addressing Institutional and Public Health Challenges

**DOI:** 10.3390/healthcare13050514

**Published:** 2025-02-27

**Authors:** Daysi Zulema Diaz-Obregón, Edgar Coila-Paricahua, Percy Soto-Becerra, César Alexander Ortiz Rojas, Alexis G. Murillo Carrasco

**Affiliations:** 1Dirección de Investigación, Instituto de Evaluación de Tecnologías en Salud e Investigación—IETSI, EsSalud, Lima 15072, Peru; daysiz.diaz.o@gmail.com (D.Z.D.-O.); edgar.coila@essalud.gob.pe (E.C.-P.); percy.soto@upn.edu.pe (P.S.-B.); 2Centro de Innovación e Investigación Traslacional en Salud, Universidad Privada del Norte, Lima 15072, Peru; 3Organization for Medical Innovation and Collaboration for Sciences—OMICS, Lima 15072, Peru; ortizr@alumni.usp.br

**Keywords:** health research priorities, health policy, public health, health services research, research agenda, health planning, evidence-based policy

## Abstract

**Background/Objectives:** Addressing health research priorities in public institutions is crucial for efficient resource allocation and policy impact. This study aims to describe the development of Peru’s Social Health Insurance (ESSALUD) 2023–2025 research portfolio, which aligns with institutional priorities and focuses on improving decision-making for population health. **Methods:** The Health Research Directorate (DIS) of ESSALUD led a structured three-phase process, engaging multidisciplinary teams and utilizing a group model-building approach to generate research ideas. Twelve working groups were established, corresponding to ESSALUD’s prioritized health topics, to identify key institutional challenges and propose research ideas. **Results:** A total of 338 research ideas were generated from 217 identified problems. These ideas were classified using the UK Health Research Classification System (HRCS) and scored based on nine dimensions to prioritize execution. Research ideas primarily focused on health services (57.7%) and disease management (16.9%). High-priority topics included cancer, mental health, malnutrition, and antimicrobial resistance. As a result of this implementation, ESSALUD resources were positively concentrated in the HRCS research activities ‘Health and social care services research’ (51.85%) and ‘Etiology’ (44.44%) for the period 2023–2025. **Conclusions:** The development of ESSALUD’s research portfolio identified key areas such as health services, health economics, and prevention, essential for evidence-based decisions and sustainability. Multidisciplinary participation ensured solutions aligned with real needs, promoting equity and continuous improvement in Peru’s health system.

## 1. Introduction

Scientific research in public health is essential for the development of countries and is expanding in low- and middle-income countries. However, it faces various challenges such as the scarcity of institutional data, inadequate funding, and the need for research outcomes to effectively support decision-making [1]. In Peru’s health sector, there has been a growing commitment to research, evidenced by the notable performance in the 2024 Scimago Ranking [2], an index that evaluates scientific output and the quality of research. In this ranking, Peru’s Social Health Insurance (ESSALUD, https://www.essalud.gob.pe/ accessed on 25 January 2025) achieved fourth place, reflecting significant progress in scientific production at the Latin American level.

Despite these achievements, there remains a substantial gap between ESSALUD’s scientific output and its declared institutional priorities. Between 2017 and 2020, only 10% of scientific articles published with ESSALUD affiliation addressed at least one health research priority, focusing primarily on public health and diabetes mellitus [3]. Furthermore, another study found that only 15.9% of publications in Peruvian health sciences journals were related to the ’National Health Research Priorities in Peru for the 2016–2021 period’ [4], raising questions about the alignment of ESSALUD’s current research with its established priorities.

The Health Technology Assessment and Research Institute (IETSI) of ESSALUD, responsible for regulating, managing, and developing research aimed at benefiting the insured population and ensuring the institution’s financial sustainability, issued Resolution No. 24-IETSI-2023 in March 2023, which sets the research priorities for the 2023–2025 period. This resolution outlines ESSALUD’s research priorities for 2023–2025, including 12 priority topics and 4 subtopics for each. For the first time, digital health is included as a research priority, marking a significant milestone [5].

Moreover, it was considered that, according to the Commission on Health Research for Development, “For the most vulnerable populations, research holds immense untapped potential to drive meaningful change and address critical inequities” [6]. This underscores the importance of focusing the health research profile of low- and middle-income countries on equity, which justifies the need to establish research priorities. Therefore, setting health research priorities is essential to efficiently allocate resources, address critical needs, influence health policies, promote equity, and foster collaboration. These actions should ensure that research is relevant and has a significant impact on improving public health, benefiting the most vulnerable populations. It also implies that research findings are used by decision-makers to formulate effective policies, enabling priorities to be adjusted and policies to be iteratively reformed, thereby closing the loop toward equity.

Despite the importance of establishing research priorities in the public sector, merely stating them is not enough. Effective implementation strategies are required to carry out research aligned with the established priorities, benefiting both the insured population and the institution. In this context, ESSALUD, through IETSI, has developed a portfolio of research ideas as part of its plan to strengthen research, which may represent a step toward translating the identified priorities into concrete actions or applied research.

This article describes the process of developing a portfolio of research ideas centered around ESSALUD’s twelve priority health research topics for the 2023–2025 period, supported by Peruvian researchers approved by the Peruvian Science and Technology Council (Researchers RENACYT), and analyzes the perspectives and challenges associated with this initiative.

This study aims to identify critical research gaps in areas fundamental to promoting equity and ensuring the sustainability of the healthcare system. By employing a systematic prioritization process and integrating multidisciplinary teams, this study goes beyond previous efforts by directly linking research priorities to the actual needs of healthcare systems. This approach ensures that research outcomes address urgent gaps, support evidence-based decision-making, and optimize resource allocation in public health.

## 2. Materials and Methods

### 2.1. Overview of the Process for Developing the Portfolio of Research Ideas

Between April and July 2023, the Health Research Directorate (DIS) of the IETSI planned and led the development of a portfolio of research ideas based on priority topics, aimed at addressing critical health problems and improving decision-making in ESSALUD. The list of prioritized topics and subtopics is detailed in Appendix A.

The working team, composed of professionals from various disciplines across the three levels of ESSALUD’s healthcare system, focused on identifying institutional issues within each of the twelve priority research topics of Peru’s Social Health Insurance. The process consisted of three phases, summarized in Figure 1.

### 2.2. Identifying the Need for a Method to Prioritize Research Development in ESSALUD

The Health Research Directorate of ESSALUD, responsible for regulating, managing, and developing research to maximize the benefit for the insured population and ensure financial sustainability, identified significant challenges in its research development. Despite having defined priority areas, between 2017 and 2020, only 10% of the scientific publications affiliated with ESSALUD addressed at least one of these priorities, primarily focusing on public health and diabetes mellitus [3]. This gap highlights that the declared priorities are too broad, allowing for approaches disconnected from the actual needs of the healthcare system.

Additionally, the predominance of clinical research limited the exploration of other necessary approaches, such as operational research, health services research, and health economics. The lack of a clear methodology to guide research development around priority topics further exacerbated this disconnection.

In response, a methodological plan was designed to strengthen research aligned with institutional priorities. This plan involved the creation of 12 interdisciplinary working groups composed of ESSALUD professionals, recognized experts, and scientists to identify and prioritize research ideas that effectively address the sector’s needs. This approach aims to bridge the gap between institutional priorities and scientific output, fostering strategic and operational alignment.

### 2.3. Formation of Working Groups for Each Research Line Declared a Priority for ESSALUD for the 2023–2025 Period

To develop the “Portfolio of Research Ideas Based on Priority Topics for the Maximum Benefit of the Insured Population and Financial Sustainability of ESSALUD 2023–2025”, an introductory meeting was held with support from the General Management and participation of Network Managers and Service Managers from representative ESSALUD Healthcare networks in Lima, Peru (Almenara, Rebagliati, and Sabogal). It was agreed to open a call for participation from professionals across the three networks in Lima and to form twelve working groups corresponding to the priority research topics for ESSALUD. Each group was composed of 6 to 9 professionals, including managers and care personnel from various professions and different levels of complexity of Health Service Providers (IPRESS) of ESSALUD. These professionals identified needs, problems, or information gaps they faced daily within the institution according to the subtopics. Each group also included researchers from the National Registry of Science, Technology, and Innovation of Peru (RENACYT) with expertise in research methodology. The aim was to transform the issues raised by the working groups into research ideas, seeking solutions to the identified problems through the generation of scientific evidence.

The following participant structure was planned for each working group:A researcher from DIS or Complementary Medicine Management, acted as the leader of the working group.One or two RENACYT researchers serve as focal points for each research line.One or two managers or professionals from an IPRESS or central level.Two managers or professionals from an IPRESS for each research line.An additional researcher was selected via an open call, meeting the requirements of being a current or former ESSALUD employee, having experience in the prioritized research topic, and being a RENACYT researcher.

The methodology for generating research ideas based on priority topics was grounded in group modeling [7]. This process involved stakeholders for each prioritized research topic, conducting a participatory workshop to understand problems, and explore their origins, contributing factors, and potential solutions or interventions. The leaders of the twelve working groups received prior training in the applied methodology, which followed a predefined eight-step program detailed below:Welcome and Introduction of Group Members: (10 min) The “spider web” dynamic was used, where the facilitator started by holding a ball of yarn, introducing themselves, and stating their expectations. The yarn was then passed to the person opposite, who repeated the process, forming a web.Presentation of Topics and Subtopics: (10 min). The facilitator or leader welcomed all participants, explained the session’s purpose, presented the working group’s topic and subtopic, and emphasized the importance of their contributions to ESSALUD’s research agenda.Brainstorming: (30 min). Participants in the groups identified specific problems related to the selected thematic field. To do this, they reviewed the topics and subtopics outlined in the 2023–2025 Health Research Priorities, ensuring that the ideas generated aligned with these guidelines. Problems were written on red paper cards measuring 15 × 10.5 cm and placed on a flip chart that was pre-divided into the four subtopics of the priority theme. Participants were free to propose as many problems as they wished, with no restrictions on quantity or need for justification. Each problem had to include two components: (a) a gap, absence, or limitation, and (b) the consequence resulting from it.Explanation and Reflection on Identified Problems: (40 min). Participants justified the identification of each problem, explaining its relevance within the context of the addressed topic. During this stage, the wording was refined, or problems were clarified as needed, ensuring accuracy and clarity in their presentation.Clustering of Identified Problems: (20 min). Participants grouped problems to form broader categories, eliminating redundancies and identifying more specific issues.Generation of Research Ideas: (40 min). After grouping the problems into categories, the group worked on generating research ideas aimed at addressing the identified issues. Blue paper cards measuring 15 × 10.5 cm were used for this purpose. Participants wrote their research ideas on the cards and placed them around the problem categories, clearly indicating which problem each idea was associated with.Review, Feedback, and Synthesis of Research Ideas: (30 min). Ideas were organized, connections identified, some reformulated, and duplicates removed.Preparation and Presentation of Research Ideas by Each Group: (45 min). Each group selected a representative to present their ideas during the plenary session.

### 2.4. Consolidation and Publication of Research Ideas

The research ideas generated were reviewed and consolidated by the DIS technical team at IETSI and published in a document entitled: “Portfolio of Research Ideas for the Maximum Benefit of the Insured Population and Financial Sustainability of ESSALUD for the 2023–2025 Period” [8].

### 2.5. Analysis and Prioritization of the Research Portfolio

After collecting the portfolio of research ideas, an analysis and proposal for prioritizing research execution at ESSALUD for 2023–2025 were conducted. The research ideas were scored using an instrument developed by DIS and validated by expert judgment, with criteria for prioritization including nine dimensions, such as study duration, affected population coverage, cost–benefit analysis, need for institutional or external support, the feasibility of resources and data, as well as the scope and innovation of the expected results (see Appendix A).

Scores for each dimension were summed (non-weighted), and the total score was categorized into tertiles for prioritization: high priority (tertile 3), medium (tertile 2), and low (tertile 1).

### 2.6. Classification of Research Ideas

Finally, the research ideas were classified according to the UK Health Research Classification System (HRCS) [9]. This widely recognized and internationally utilized system provides a standardized framework for classifying and analyzing health research. Its multidimensional approach combines health categories and research activities, enabling the prioritization of key areas based on the health system’s needs. This system facilitates the identification of gaps, comparison of research portfolios, and international collaboration, promoting evidence-based decisions aligned with strategic objectives of equity and sustainability [10].

The HRCS scoring criteria were developed through consultations with health research experts and the application of principles of consistency and strategic relevance. Validation was achieved through comparative analyses across multiple organizations, demonstrating its capacity to classify research accurately and reproducibly. Furthermore, its implementation in countries such as the United Kingdom, Sweden, and Canada has confirmed its practical utility for prioritization and strategic decision-making in health research [10].

In this study, two researchers conducted the classification based on the proposed system, resolving discrepancies by consensus.

### 2.7. Evaluation of the Impact of the Implementation of This Health Research Strategy

We retrieved data from the technical and research reports from ESSALUD, located in their Institutional Report System (https://ietsi.essalud.gob.pe/investigaciones-realizadas/ accessed on 5 February 2025). All reports between 2018 and 2025 were analyzed and categorized according to the HCRS Research activities [9] and the year of publication. After we divided three categories regarding the year of publication, we considered 2018–2019 (before developing the Health Research Strategy), 2020–2022 (pandemics and developing of the Health Research Strategy), and 2023–2025 (after application of the Health Research Strategy). Finally, bar plots showing the percentage distribution of HCRS research activities into year-based categories were produced using R software v.4.4.0.

## 3. Results

### 3.1. Characteristics of the Participants

On 20 July 2023, the process for generating ESSALUD’s research portfolio involved a total of 99 participants, carefully selected to represent a wide range of expertise and perspectives. The participants involved in the research portfolio development process represented a diverse array of professional backgrounds, reflecting the multidisciplinary nature of the initiative. Physicians formed the majority, accounting for 67 participants (67.7%), followed by nurses and midwives, each contributing five participants (5.1%). Economists and engineers were also well represented, with four participants each (4.0%). Other professions, including psychologists (three participants, 3.0%), medical technologists (three participants, 3.0%), administrators (two participants, 2.0%), and pharmaceutical chemists (two participants, 2.0%), provided valuable perspectives. Additionally, one representative from the fields of law, biology, nutrition, and dentistry contributed to the process (one participant, 1.0% each). This diverse composition highlights the inclusion of varied expertise, ensuring a comprehensive and interdisciplinary approach to addressing the institution’s research priorities.

Among these participants, 87 were directly assigned to 12 working groups corresponding to ESSALUD’s priority health topics for the 2023–2025 period. An additional 12 members from the Health Research Directorate (DIS) of the Health Technology Assessment and Research Institute (IETSI) and the Complementary Medicine Management acted as group leaders. The group leaders were all certified researchers registered in the National Registry of Science, Technology, and Innovation of Peru (RENACYT), ensuring methodological rigor and subject matter expertise.

The participants included healthcare managers, professionals, and researchers from various levels of ESSALUD’s healthcare system. Notably, there was a balanced gender distribution, with males accounting for 52.75% and females for 47.25%. The professional composition reflected diverse institutional roles and levels of care:Primary care (Level I): 18.2% of participantsSecondary care (Level II): 10.1% of participantsTertiary care (Level III): 16.2% of participants

The remaining participants (42.4%) were researchers, with 34 out of 42 being RENACYT-certified. This high proportion of certified researchers underscores the emphasis placed on evidence-based approaches in developing the research portfolio (Table 1). One RENACYT researcher simultaneously held a managerial position and represented a Level I facility, reflecting the integration of clinical and academic expertise in decision-making processes.

Participation was geographically diverse, encompassing professionals from ESSALUD’s Rebagliati, Almenara, and Sabogal networks, as well as its Central Office. The inclusion of representatives from facilities of varying complexity levels (primary, secondary, tertiary) ensured that the process captured a comprehensive understanding of institutional needs and challenges.

In each working group, members focused on identifying problems, gaps, or needs related to their assigned research topic. Key priority areas addressed included high-burden diseases, such as cancer, mental health, cardiovascular diseases, and maternal health, as well as system-wide challenges like service provision, digital health, and antimicrobial resistance. The collaborative structure of the working groups facilitated interdisciplinary discussions, combining practical insights from healthcare providers with analytical inputs from researchers.

This multidisciplinary and multi-level representation was critical in ensuring that the proposed research ideas were not only scientifically robust but also practically relevant to ESSALUD’s operational and strategic goals. By incorporating diverse professional experiences and leveraging the expertise of RENACYT-certified researchers, the initiative created a solid foundation for addressing pressing public health challenges through actionable and impactful research projects.

### 3.2. Proposed Research Problems and Ideas

Each workgroup identified the issues associated with their prioritized research line, leading to the presentation of 217 problems, which resulted in a total of 338 research ideas (Table 2). These ideas are intended to develop research projects and generate scientific evidence to optimize decision-making. The research portfolio was included in the DIS Research Work Plan for the 2023–2025 period at ESSALUD.

The prioritization process classified the 338 research ideas into three levels: high priority (Tertile 3), focusing on studies with significant public health impact, such as cancer, mental health, and service management; medium priority (Tertile 2), with relevant but moderately impactful research; and low priority (Tertile 1), addressing less urgent projects or those with greater implementation challenges. The classification was based on nine key dimensions, including population coverage, cost–benefit analysis, and resource feasibility (Table 3). This system strategically allocates resources to research with the highest impact potential, aligning outcomes with ESSALUD’s critical needs and ensuring the generation of evidence to support institutional decision-making.

The highest scores were observed in Antimicrobial Resistance, Malnutrition and Anemia, Resource Generation and Funding, Service Provision, and Cancer (Figure 2).

According to the Health Research Classification System (HRCS) [9], 57.7% (195 of 338) of the ideas focused on research related to healthcare and social services, covering areas such as service organization, health economics, well-being, policy, ethics, and research governance. This type of research was more frequently required across five priority areas: cancer, COVID-19, tuberculosis and other infectious diseases, antimicrobial resistance, maternal, perinatal and neonatal health, and mental health.

The second most requested group, comprising 16.9% (57 of 338), involved research on clinical and individualized disease management. In third place were etiology studies with 10.1% (34 of 338), aiming to identify and characterize determinants contributing to the cause, risk, and development of diseases or conditions.

After grouping diseases into a single category (excluding management topics), nearly 50% of the research ideas pertained to health services research. The remaining percentage was distributed among Disease Management, Etiology, Detection, Treatment Evaluation, and Prevention (Figure 3, panel left). More than 80% of the management-related research ideas focused on Health Services, followed by Disease Management and Etiology (Figure 3, panel right).

### 3.3. Impact of Implementing This Health Research Strategy

After analyzing all the institutional research reports produced for ESSALUD professionals from 2018 to 2025, we identified a total of 99 studies, distributed as follows: 21 studies from 2018 to 2019 (the period preceding the implementation of the health research strategy), 51 studies from 2020 to 2022 (a period encompassing the COVID-19 pandemic and the development phase of the health research strategy), and 27 studies from 2023 to 2025 (reflecting the outcomes of the strategy’s implementation). Figure 4 presents the HCRS research categories represented by these studies.

In the 2018–2019 period, research activities were distributed across multiple categories, with Health and Social Care Services Research (33.33%) being the most prominent, followed closely by Etiology (28.57%). Nevertheless, research projects before implementation included research related to the prevention of diseases and conditions, promotion of well-being and detection, screening, and diagnosis, each equally representing 14.29%, while underpinning research accounted for a smaller portion (9.52%). Notably, there was no recorded research on the evaluation of treatments and therapeutic interventions. Moving into 2020–2022, a significant shift was observed, with Etiology increasing to 33.33%, overtaking Health and Social Care Services Research, which declined to 17.65%. The category detection, screening, and diagnosis also gained prominence, rising to 15.69%, while prevention of disease and conditions, and promotion of well-being remained relatively stable at 13.73%.

Nevertheless, the most relevant changes occurred in 2023–2025, where Health and Social Care Services Research saw a substantial surge, reaching 51.85%, making it the dominant research focus. Etiology also continued to grow, rising further to 44.44%. This pattern highlights a shifting emphasis toward broader systemic health and social care research, coupled with a sustained focus on disease causation (etiology), while areas like screening, prevention, and therapeutic evaluation have declined in prominence.

Institutional reports published between 2023 and 2025, have analyzed alternatives to improve ESSALUD healthcare and service. Some of these studies aimed to detect inequities and propose potential solutions in different specialties and diseases. Here, we review some interesting studies (RRI-07-2024, RRI-10-2023, RRI-09-2023, and RRI-02-2023) that resulted from implementing this Health Research Strategy.

The document RRI-07-2024 (https://ietsi.essalud.gob.pe/wp-content/uploads/2024/10/RRI-07-2024-brechas.pdf accessed on 22 January 2025) examines inequities in Peru’s health system by identifying gaps in access, quality, and distribution of healthcare services. It highlights disparities based on socioeconomic status, geographic location, and healthcare infrastructure, demonstrating how marginalized populations—especially those in rural and indigenous communities—face systemic barriers to care. The study employs a rights-based and evidence-driven approach, advocating for policy changes that promote equitable healthcare delivery. By analyzing structural weaknesses and proposing strategies for improvement, the research underscores the need for inclusive health policies that ensure universal access and reduce inequalities in the Peruvian healthcare system.

In addition, the document RRI-10-2023 addresses inequities in the Peruvian healthcare system by evaluating the high prevalence of postpartum anemia among women treated in ESSALUD hospitals. The findings reveal that factors such as young maternal age, limited access to prenatal care, significant blood loss during childbirth, and routine episiotomies are associated with a higher incidence of postpartum anemia. By analyzing these factors in the context of socioeconomic disparities and unequal access to preventive healthcare services, the research emphasizes the urgent need for improved screening and treatment strategies for postpartum anemia. It advocates for more humanized and inclusive maternal healthcare policies to reduce disparities and enhance maternal health outcomes in Peru.

Approaching neoplasies, the RRI-09-2023 evaluated cancer prevalence and mortality within the ESSALUD-insured population. The research reveals significant geographic variations, with higher cancer prevalence observed in the eastern coastal areas and the northeastern Amazonian region of Loreto, suggesting unequal access to early diagnosis and treatment. The study emphasizes that while breast and prostate cancer remain the most common types, the rising incidence of cervical and colorectal cancer signals gaps in prevention and screening programs. Moreover, the findings indicate that the implementation of electronic health records in 2019 improved data collection but also revealed underreporting issues, particularly in underserved regions. The study underscores the urgent need for targeted interventions, enhanced screening efforts, and equitable healthcare distribution to mitigate regional disparities and improve cancer care outcomes across Peru.

Regarding patient satisfaction with health services, the RRI-02-2023 assessed this topic in the context of a specialized rehabilitation service. The study revealed that while overall satisfaction was 72.4%, key aspects such as responsiveness (58.1%) and reliability (68.8%) scored lower, indicating gaps in service efficiency and trust. These findings suggest that despite the perceived safety and empathy in care, patients with disabilities face challenges in timely and effective rehabilitation services. The study proposed strategies to improve healthcare accessibility, responsiveness, and infrastructure to ensure equitable, high-quality rehabilitation care for all patients, particularly those in vulnerable populations.

Thus, the implementation of this health research strategy is yielding some results through the reassignment of priorities and the management of projects that offer improvements to the current system for Peruvians with social benefits in ESSALUD, which could later be expanded to other healthcare service networks in the country.

## 4. Discussion

Establishing research priorities in health is crucial for efficient resource allocation, addressing critical needs, influencing health policies, promoting equity, and fostering collaboration among various health system stakeholders [6]. Thus, ESSALUD developed a systematic and planned process for implementing these prioritized investigations for the priorities in 2023–2025. This process involved interdisciplinary participation from institutional professionals, including RENACYT researchers, managers, and health professionals across all three levels of care, resulting in a portfolio of 338 health research ideas addressing defined priority topics.

A group model-building methodology was employed to generate this health research portfolio [7], facilitating the implementation of prioritized studies within a public institution like ESSALUD. This approach fostered the participation of multidisciplinary teams to ensure equity in decision-making and effectively address the complexities of the health system. These teams comprised healthcare professionals from all levels of care (primary, secondary, and tertiary) and research scientists specializing in the various prioritized research lines.

The role of healthcare personnel was to provide key insights into the specific challenges of each level, such as disease prevention at the primary level and the management of chronic and complex diseases at higher levels. Researchers analyzed these problems scientifically, fostering collaborative dialog to propose methodologically robust and practical studies that address identified needs or generate valuable evidence for decision-making.

This approach promotes targeted, evidence-based solutions for all levels of care, addressing the often-overlooked needs of primary care and alleviating economic and quality-of-life burdens associated with chronic diseases. Such multidisciplinary collaboration strengthens the health system’s ability to implement equitable, sustainable, and institution-specific evidence-based policies and practices.

Group model-building emerges as an innovative method for addressing complex and long-standing public health issues that often defy traditional intervention approaches. This model engages stakeholders in participatory workshops to explore perceptions of a problem’s origin, contributing factors, and potential solutions or interventions. The approach has been widely used in other contexts, such as business and criminal justice, and its application in public health, characterized by dynamic complexity, has also been successful [7]. A systematic review by Estrada-Magbanua et al. (2023) highlighted the effectiveness of this model, examining 72 studies published between 2002 and 2022. Findings indicate that the frequent application of this approach in the United States and Australia is associated with tangible benefits and solutions for both society and institutions. These results support the utility and relevance of group model-building as a valuable tool in the arsenal of public health strategies for addressing complex and multifaceted challenges [7].

Once the research ideas portfolio was created, the challenge arose to determine which studies to prioritize, as all proposals were important. Decisions had to be made regarding which would be implemented in the short, medium, or long term. To address this challenge, an instrument was developed to facilitate the distribution of research ideas according to priority level, classifying them as high, medium, or low. This instrument assessed various dimensions, such as the time required to execute each study, the coverage of the affected population, cost–benefit analysis, the need for internal or external collaboration, resource availability, and the potential impact of the results. These dimensions allowed for an objective and systematic selection of studies to be conducted, thereby minimizing the influence of subjectivity or individual interests in the selection process. This instrument underscores the importance of collaboration with intra- and extra-institutional networks, including agreements with universities, to effectively and efficiently advance the prioritized studies in the portfolio.

The results indicated that the research priority topics focused on cancer, mental health, malnutrition and anemia, antibacterial resistance, and health service management. These areas encompassed the highest number of ideas classified as high priority according to the instrument’s evaluation.

Following the generation of ESSALUD’s portfolio of 338 research ideas, it became necessary to identify the corresponding research activity for each idea using the Health Research Classification System (HRCS) [9]. This internationally recognized system provides a widely accepted framework for categorizing health research [10]. The results offer a comprehensive understanding of the current needs within ESSALUD’s health system.

The impact of the strategy implemented for the creation of the Health Research Portfolio, based on priority topics for the Peruvian Social Health Insurance (ESSALUD) during the 2023–2025 period, is reflected in the scientific output of the Institute for Health Technology Assessment and Research (IETSI). In this regard, a significant shift in the research approach adopted by this public institution is observed, with a notable emphasis on health services research and disease management, accounting for 51.85% of all research projects. This percentage contrasts with the 17.65% recorded during the 2020–2022 period and the 33.33% reported for the 2018–2019 period.

According to our study, the development of this type of research has a considerable institutional impact, as it generates valuable evidence for decision-making processes that affect all patients affiliated with ESSALUD. This effort significantly enhances the relevance and applicability of research conducted by ESSALUD, as it not only focuses on the country’s priority health issues but also facilitates the identification of effective solutions to healthcare needs and challenges.

In this context, multidisciplinary studies have promoted a preventive approach, transforming healthcare delivery. This strategic approach, supported by the evidence generated through prioritized research, aims to transition from a reactive model to a proactive one, emphasizing prevention.

The classification of research ideas according to HRCS revealed that more than 50% of the required research activity focuses on health and social services. This finding aligns with the growing demand for attention in this area and highlights the insufficient supply of health services nationally, evidencing a significant gap in the health system.

Within the healthcare and social services research group, studies in health economics and well-being are crucial for guiding strategic decisions and ensuring financial sustainability in the public sector. Although ESSALUD does not have a dedicated unit for this field, the need for such studies is evident [11]. Health economic evaluations, especially concerning healthcare technology acquisitions, are essential for making efficient and equitable decisions. In our country, these evaluations focus on the effectiveness and safety of technologies, with little emphasis on cost–benefit and cost-utility analyses, potentially leading to partial decisions and risks of institutional underfunding [11,12,13].

Furthermore, health services research is critical for addressing the inherent complexities of providing medical care and improving care processes. Health systems, particularly in our context, face significant challenges related to accessibility, quality of care, and equity in health services. This field of research offers the opportunity to identify barriers, evaluate interventions, plan resources, and make strategic public health decisions. It also allows for the development of effective solutions to improve medical service provision and, ultimately, promote population health [14]. The current availability of electronic health records at ESSALUD allows for rapid national-level data access, enabling such research with a significant impact on institutional decision-making.

These results align with the literature, highlighting that one of the main types of research required in complex organizations, such as health systems, is health services research. The outcomes of such research provide scientific evidence that can be used to design more effective and equitable health policies and programs [15]. This is especially relevant in a context where resources are limited and need to be maximized.

One of the main challenges facing the health sector is reducing long waiting times, particularly for priority conditions such as cancer, where effective oncology care is critical. Addressing this challenge requires a thorough analysis, identification of areas for improvement, and the implementation of continuously improved management practices [16]. For example, in the care process for priority cancer patients, such as those with breast, prostate, and cervical cancer, ESSALUD has recorded an average waiting time of four months from diagnosis to the start of treatment [17].

The second most frequent group included research categorized as “disease and condition management”, comprising 16.9% of the research ideas. These studies play a crucial role in improving individual healthcare, developing effective interventions, and advancing scientific knowledge in medicine. Their direct impact on patient quality of life and clinical practice makes them a priority in the health research agenda.

Less frequently, HRCS classifications included research on etiology, diagnosis, treatment evaluation, disease prevention, and basic research. Some of these studies are being conducted by the pharmaceutical industry, with Soto-Ordoñez (2022) reporting that 94.9% of ongoing clinical trials at ESSALUD were industry-sponsored, with 36% related to oncology [18]. Research primarily focuses on neoplastic diseases, musculoskeletal disorders, non-communicable diseases, and chronic respiratory conditions, representing 59% of the total [19].

As in this study, some group model-building (GMB) proposals have emerged as a participatory tool to tackle complex challenges in healthcare systems. Recent studies highlight its potential for driving transformative change by fostering collaboration among diverse stakeholders and addressing long-standing healthcare issues.

The study by Forrester-Bowling et al. (2024) demonstrated the adaptability of GMB in mental healthcare settings, showcasing its alignment with co-design principles and person-centered care. This participatory approach facilitated meaningful collaboration with stakeholders, including individuals with lived experiences, highlighting its utility in redesigning mental health services to better meet patient needs [20]. Similarly, the systematic review by Estrada-Magbanua et al. (2023) underscores the wide-ranging applications of GMB in public health and healthcare research. This work emphasizes the role of GMB in implementing evidence-based interventions and policies, addressing the limitations of traditional intervention models when dealing with multifaceted public health challenges [7].

Another significant application of GMB is in addressing disparities in healthcare, as we intend to do in Peruvian Social Health Insurance (ESSALUD). The study by Gillani et al. (2024) employs system dynamics and GMB to explore healthcare access barriers for sexual and gender minority populations. Their findings underscore GMB’s value in visualizing complex systems and fostering targeted interventions to reduce disparities [21]. Furthermore, Munce et al. (2024) highlight the critical role of GMB in consensus-building initiatives, particularly in engaging patients, caregivers, and diverse knowledge users. This approach ensures that healthcare solutions are inclusive and grounded in stakeholder perspectives [22].

Overall, these studies affirm the transformative potential of GMB in healthcare, reinforcing our purpose to implement it in ESSALUD assistance and research mission. It not only addresses system complexities but also bridges gaps between research and practice. The consistent emphasis on stakeholder engagement highlights a paradigm shift toward participatory and system-thinking approaches, paving the way for sustainable and equitable healthcare solutions. Future research could explore the integration of GMB with digital tools and its application across diverse cultural and organizational contexts to maximize its impact.

The implementation of research priorities at ESSALUD faces several challenges, one of the main ones being the active participation of representatives from all levels of care, with an emphasis on primary care. This level is crucial for health promotion and disease prevention but is often underrepresented, making it difficult to align priorities with the system’s real needs. Additionally, limited technical capacity to conduct studies in areas such as health economics and health services, coupled with financial constraints and the need for interinstitutional collaboration, complicates the efficient execution of the research portfolio.

To address these barriers, several strategies are proposed. First, the integration of multidisciplinary teams, including managers, decision-makers, and certified researchers, must be strengthened to ensure that research priorities are aligned with institutional needs and supported by solid evidence. Second, developing training programs in operational research and health economics, as well as establishing strategic collaborations with universities and specialized research centers, would accelerate the execution of priority studies.

Furthermore, creating monitoring and evaluation mechanisms would allow for the early identification of obstacles and timely adjustments to strategies. Finally, prioritizing research on prevention and health promotion, with a significant impact on reducing the burden of chronic diseases and healthcare costs, would help close critical gaps in service provision, optimize resources, and improve patients’ quality of life. These measures would ensure the sustainability of the research portfolio and the continuous improvement of the health system.

## 5. Conclusions

In conclusion, the development of the ESSALUD research portfolio enabled the identification and prioritization of critical areas such as health services, health economics, and well-being, which are fundamental for evidence-based decision-making and the sustainability of the health system. The participation of multidisciplinary teams ensured solutions aligned with the real needs of the health system. However, the need to strengthen research in prevention and health promotion was identified, as these areas are key to reducing the burden of diseases and improving quality of life. This approach fosters equity, sustainability, and the continuous improvement of the health system in Peru.

## Figures and Tables

**Figure 1 healthcare-13-00514-f001:**
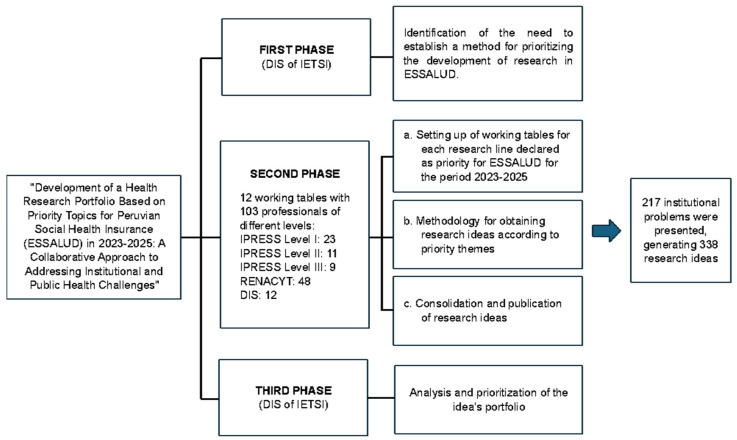
Overview of the research portfolio development process in ESSALUD for the period 2023–2025. This figure outlines the structured methodology implemented by ESSALUD’s Health Research Directorate to develop a research portfolio aligned with institutional priorities for 2023–2025. The process consists of three phases: (1) Identifying research needs and gaps through multidisciplinary engagement, (2) forming and training 12 working groups corresponding to ESSALUD’s priority health topics, and (3) analyzing and prioritizing the research ideas generated. The figure highlights key entities involved, including the Health Technology Assessment and Research Institute (IETSI), Health Service Providers (IPRESS), and the National Registry of Science, Technology, and Innovation of Peru (RENACYT).

**Figure 2 healthcare-13-00514-f002:**
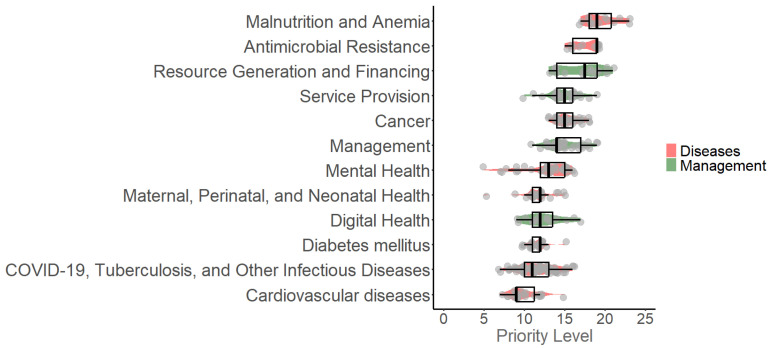
Distribution of priority levels across research topics in the ESSALUD portfolio. This figure presents a boxplot visualization of research topics categorized by priority level based on a comprehensive evaluation. Priority levels were determined using nine dimensions, including public health impact, feasibility, and innovation. Topics are divided into disease-related areas (orange) and management-related topics (green), emphasizing ESSALUD’s focus on areas such as cancer, mental health, and health service management. The boxplots provide insights into the distribution of scores, helping stakeholders prioritize research investments effectively.

**Figure 3 healthcare-13-00514-f003:**
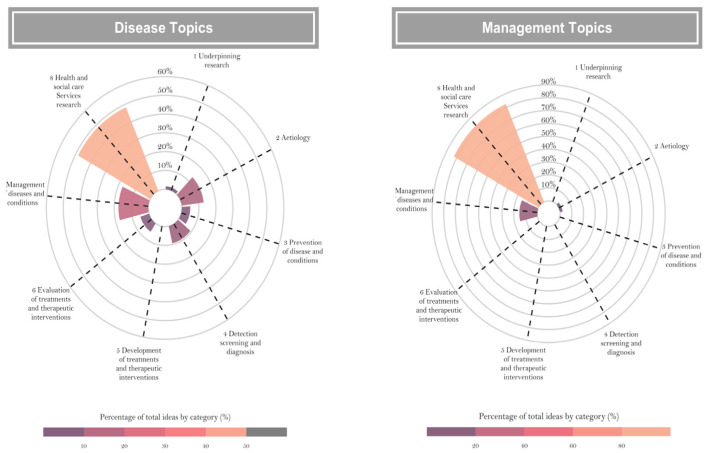
Activity group distribution for research ideas categorized by topic type. This radar chart compares the proportion of research ideas across different health research activities, as defined by the Health Research Classification System (HRCS). The left panel focuses on disease-related topics, showing a significant emphasis on health services and disease management research. The right panel highlights management-related topics, with health services research dominating the distribution. The charts provide a comprehensive overview of how research activities align with ESSALUD’s strategic goals for evidence-based improvements in public health and system management.

**Figure 4 healthcare-13-00514-f004:**
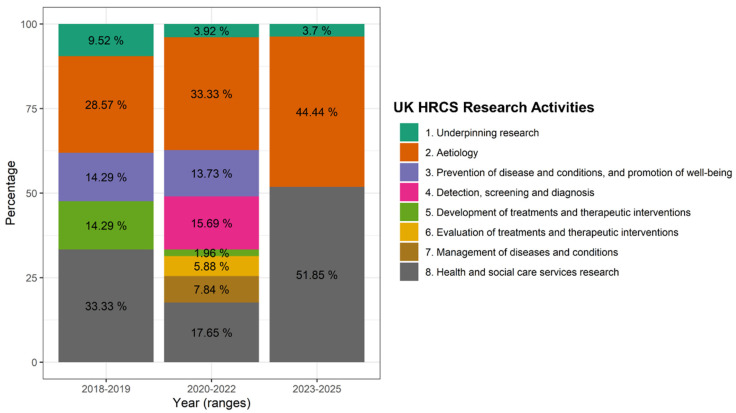
Trends in UK Health Research Classification System (HRCS) between 2018–2025. This stacked bar chart represents the percentage distribution of different Health and Care Research Activities (HCRS) across three time periods: 2018–2019 (before developing the Health Research Strategy), 2020–2022 (pandemics and developing of the Health Research Strategy), and 2023–2025 (after application of the Health Research Strategy). Each color corresponds to a specific research activity, as detailed in the legend on the right.

**Table 1 healthcare-13-00514-t001:** Characteristics of participants in the development of ESSALUD’s research portfolio.

	Topic	I	II	III	Peruvian Ministry of Health	Core ESSALUD	Researcher RENACYT		Total
							No	Yes	
1	Cancer	2	1				1	4	8
2	Mental Health	2		1			1	3	7
3	Cardiovascular Disease	1		1			1	5	8
4	Diabetes Mellitus	1	1	1			2	2	7
5	Malnutrition and Anemia	1	1	2			0	2	6
6	Maternal, Perinatal, and Neonatal Health	1	1	1			1	4	8
7	Antimicrobial Resistance	1	1	3			0	3	8
8	COVID-19, Tuberculosis, and Other Infectious Diseases	3		1			2	2	8
9	Resource Generation and Funding	1	1	1		6	0	2	11
10	Service Provision	3	2			2	0	1	8
11	Management	1	1	3		1	1	2	9
12	Digital Health	1	1	2	2	2	0	3	11
	Total of Participants	18	10	16	2	11	9	33	99
	Percentage	18.2%	10.1%	16.2%	2.0%	11.1%	9.1%	33.3%	100%

**Table 2 healthcare-13-00514-t002:** Problems identified and research ideas generated for ESSALUD’s priority health topics.

N°	Topic	N° of Identified Problems (Research Questions)	N° of Research Proposals
**Disease-related topics**			
1	Cancer	13	37
2	Mental Health	18	37
3	Cardiovascular Disease	16	24
4	Diabetes Mellitus	14	23
5	Malnutrition and Anemia	11	14
6	Maternal, Perinatal, and Neonatal Health	18	24
7	Antimicrobial Resistance	29	13
8	COVID-19, Tuberculosis, and Other Infectious Diseases	37	74
**Health system-related topics**			
9	Resource Generation and Funding	23	16
10	Service Provision	23	25
11	Management	9	36
12	Digital Health	6	15
	Total	217	338

**Table 3 healthcare-13-00514-t003:** Distribution of research ideas by priority level and health topic in ESSALUD’s research portfolio.

Priority Topics	Priority Level			Total
	High (Tertile 3)	Medium (Tertile 2)	Low (Tertile 1)	
Cancer	27	10	0	37
Mental Health	14	14	9	37
Malnutrition and Anemia	14	0	0	14
Antimicrobial Resistance	13	0	0	13
Management	12	22	2	36
Service Provision	10	12	3	25
Resource Generation and Funding	10	6	0	16
COVID-19, Tuberculosis, and Other Infectious Diseases	7	20	47	74
Maternal, Perinatal, and Neonatal Health	2	7	15	24
Digital Health	2	5	8	15
Diabetes Mellitus	1	12	10	23
Cardiovascular Diseases	1	5	18	24
Total	113	113	112	338

## Data Availability

The original contributions presented in the study are included in the article/Appendix A, further inquiries can be directed to the corresponding author/s.

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
