# Peer review of "Development of a Health Research Portfolio Based on Priority Topics for Peruvian Social Health Insurance (ESSALUD) in 2023–2025: A Collaborative Approach to Addressing Institutional and Public Health Challenges"

_healthcare, 2025, doi:10.3390/healthcare13050514_

Round 1

Reviewer 1 Report

Comments and Suggestions for Authors

Dear Authors,

Thank you for the opportunity to review your manuscript titled "Development of a Health Research Portfolio Based on Priority Topics for Peruvian Social Health Insurance (ESSALUD) in 2023-2025: A Collaborative Approach to Addressing Institutional and Public Health Challenges." I commend the effort put into this important and timely research. Below, I provide detailed feedback to help strengthen the manuscript further:

General Feedback

1.      Importance and Relevance: The manuscript addresses an essential topic—prioritizing research to optimize healthcare services and resource allocation. This focus is particularly relevant given the challenges faced by health systems in low- and middle-income countries.

2.      Methodological Rigor: The structured approach using group model-building and the prioritization framework is well-detailed and innovative.

Abstract

1.      The abstract effectively summarizes the study. However, consider explicitly mentioning the impact or anticipated outcomes of implementing this portfolio.

Introduction

1.      The introduction is well-written and provides sufficient background. However, the research gap could be more explicitly stated.

o   Suggested addition: Clarify how this study advances knowledge beyond previous efforts in aligning research priorities with health system needs.

Methods

1.      The multi-phase process is detailed and easy to follow. However, some clarification could enhance its comprehensibility:

o   Provide a brief justification for using the UK Health Research Classification System (HRCS) for prioritization.

o   Elaborate on how scoring criteria were developed and validated.

Results

1.      The results are comprehensive and well-structured, but a more detailed explanation of the prioritization outcomes (e.g., tertile categories) would add clarity.

2.      Figures and tables are informative, but Figure 2 could benefit from clearer labels or explanations for non-specialist readers.

Discussion

1.      The discussion connects findings to broader implications effectively. However:

o   Include examples of how other institutions or countries have successfully implemented similar research portfolios.

o   Elaborate on the potential barriers to implementing the proposed research ideas and how these might be mitigated.

2.      The emphasis on multidisciplinary collaboration and equity is commendable, but consider expanding on how these principles can be sustained over time.

Conclusion

1.      The conclusion provides a strong summary but could include specific, actionable recommendations for policymakers or researchers.

Additional Recommendations

1.      Citation correction: Ensure that the use of right bracket citations (e.g., "This is the right [1]." rather than (1).

2.      Recent References: Consider including more recent studies to reinforce the relevance of your findings. For example, recent literature on the use of group model-building in healthcare could strengthen your discussion.

3.      Language and Style: The manuscript is well-written but could benefit from a final round of editing to ensure consistency in terminology and grammar.

Author Response

Please verify our responses to reviewer 1 in the attached file.

Reviewer 2 Report

Comments and Suggestions for Authors

Author Response

Please verify our responses to reviewer 2 in the attached file.

Reviewer 3 Report

Comments and Suggestions for Authors

Thanks for submitting the manuscript to the journal, please find the following comments for your considerations:

1) line 16, is ESSALUD the formal short form for Peru's Social Health Insurance as seems better as PSHI for most readers.

2) line 58-59, the quoted sentence is not clear about the meaning. Would be better to paraphase the sentence.

3) line 130, please indicate the nature of types of health care professionals such as doctors, nurses, allied health

4) line 195, Table S2, where is it? State refer to supplementary materials

5) line 198-199, refer the tertile to table 3 

6) line 204, was chosen by this project ....

7) line 251, and table 3, please confirm Funding or Financing

8) line 288, 289, where are Figure 3A and 3B?

9) line 301, I am a bit confused about your project duration. Was it the study performed in 2023 for the priorities in 2023-2025. If so, it is not during the 2023-2025 period

10) line 347-348, give citation of source of this point 

11) line 434, any lessons for other countries?

Author Response

Please verify our responses to reviewer 3 in the attached file.
